# Molecular and Serological Detection of *Leishmania* spp. in Mediterranean Wild Carnivores and Feral Cats: Implications for Wildlife Health and One Health Surveillance

**DOI:** 10.3390/ani15182751

**Published:** 2025-09-20

**Authors:** Francesca Suita, Víctor Lizana, Jordi Aguiló-Gisbert, Jordi López-Ramon, João Torres Da Silva, Eduardo A. Díaz, Jesús Cardells

**Affiliations:** 1Servicio de Análisis, Investigación, Gestión de Animales Silvestres (SAIGAS), Veterinary Faculty, Universidad Cardenal Herrera-CEU, CEU Universities, 46115 Valencia, Spain; francesca.suita3@uchceu.es (F.S.); victor.lizana@uchceu.es (V.L.); jordi.aguilo@uchceu.es (J.A.-G.); jordi.lopez1@uchceu.es (J.L.-R.); joaopedro.torressilva@alumnos.uchceu.es (J.T.D.S.); 2Escuela de Medicina Veterinaria, Colegio de Ciencias de la Salud, Universidad San Francisco de Quito (USFQ), Quito 170901, Ecuador; eadiaz@usfq.edu.ec

**Keywords:** *Leishmania* spp., molecular detection, serology, One Health, reservoir hosts, wild carnivores, zoonotic disease

## Abstract

Leishmaniasis is a zoonotic disease caused by protozoa of the genus *Leishmania*, transmitted by sandflies. While domestic dogs are considered the main hosts in the Mediterranean region, several wild carnivore species may also harbor the parasite, potentially playing a role in its transmission cycle. This study assessed the presence of *Leishmania* spp. in 216 wild carnivores and 34 feral cats from the Valencian Community (eastern Spain), using both molecular (qPCR) and serological (ELISA) methods. Parasite DNA was detected in 14 individuals from five species, including red fox (*Vulpes vulpes*), Eurasian badger (*Meles meles*), American mink (*Neogale vison*), stone marten (*Martes foina*) and cat (*Felis catus*). Most animals showed no apparent signs of disease and had low parasite loads. Only one *V. vulpes*—positive by both qPCR and ELISA—showed severe skin lesions and was confirmed to be affected by sarcoptic mange, raising questions about the possible contribution of *Leishmania* to disease severity. These findings support the hypothesis that wildlife may act as silent reservoirs and highlight the need for further research on their epidemiological role within a One Health framework.

## 1. Introduction

Leishmaniasis is one of the most important emerging zoonotic diseases in several parts of the world, caused by protozoan parasites of the genus *Leishmania* and transmitted by phlebotomine sandflies [1]. Among these, *L. infantum* is the primary agent of zoonotic visceral leishmaniasis in the Mediterranean basin, the Middle East, and South America, where it remains a major public health concern [2,3].

In endemic regions, domestic dogs are considered the main reservoir host [4], and the disease can be fatal if left untreated [5]. However, domestic cats are also considered reservoirs [6], and increasing evidence suggests that several wild carnivore species—including red foxes (*Vulpes vulpes*), grey wolves (*Canis lupus*), Eurasian badgers (*Meles meles*), lynxes (*Lynx* spp.), American minks (*Neogale vison*) and martens (*Martes* spp.)—may also harbor and suffer the parasite, contributing to its maintenance in natural ecosystems [7,8,9]. Among them, red foxes have been known to carry the parasite since at least 1987 [10], likely due to their abundance, adaptability to peri-urban environments [11], and ecological overlap with domestic dogs.

The involvement of wildlife reservoirs in transmission cycles complicates control strategies focused solely on domestic animals and highlights the need for multidisciplinary, One Health-based approaches to zoonotic disease surveillance and management [2]. Moreover, infection in wild species, including endangered carnivores, raises conservation concerns, as leishmaniasis may pose an additional threat to vulnerable populations [8]. Of particular concern is the infection of endangered species, such as the Iberian lynx (*Lynx pardinus*), where *L. infantum* has been detected using both molecular and serological methods [12]. Although these findings raise relevant conservation and health management considerations, the actual clinical impact of *Leishmania* infection on the Iberian lynx remains uncertain and requires further longitudinal studies [12].

The geographic range of *L. infantum* is also expanding, driven in part by climate change, which affects both the distribution of sandfly vectors and the dynamics of parasite transmission [13]. Previously non-endemic areas in northern Spain and Italy have reported increasing numbers of clinical cases and vector presence, with environmental changes believed to facilitate this shift [14]. In addition, the movement of infected dogs from endemic to non-endemic areas contributes to the introduction of the parasite into new regions [2].

In southeastern Spain, high prevalence rates of *L. infantum* have been reported in wild carnivores. For example, Risueño et al. detected the parasite in 44.9% of foxes sampled in the Region of Murcia and the Valencian Community using real-time PCR on tissue samples [5]. However, their study included a limited number of individuals from the Valencian Community, leaving gaps in our understanding of local epidemiology.

Building upon these findings, the present study expands the sample size and includes a broader range of mesocarnivore species to provide a more comprehensive assessment of *L. infantum* circulation in wildlife across the Valencian Community. Given their abundance, outdoor lifestyle, and lack of sanitary control, feral cats and wild carnivores may act as potential primary or secondary reservoirs of the parasite. The aim of this study was to investigate their role in the epidemiology of leishmaniasis in the endemic region of the Valencian Community using both molecular (qPCR) and serological (ELISA) diagnostic techniques.

## 2. Materials and Methods

### 2.1. Study Area

The study was conducted in the Valencian Community, an autonomous region on the eastern coast of Spain, characterized by a Mediterranean climate with hot, dry summers and mild, wet winters [15]. These climatic conditions create optimal habitats for phlebotomine sandflies [16,17,18]. The region encompasses diverse ecosystems, including coastal zones, wetlands, and sub-Mediterranean forests, which influence both sandfly distribution and the abundance of potential reservoir hosts [19,20]. As an endemic area for leishmaniasis, the Valencian Community has reported an increasing number of human and animal cases in recent years [21]. Climate-related factors, such as rising temperatures and altered precipitation patterns, are expected to impact vector ecology and potentially shift transmission dynamics [22,23]. The study area included multiple locations representing different ecological settings, allowing for a broad assessment of *Leishmania* spp. presence in local mesocarnivore populations.

### 2.2. Sample Collection and Passive Surveillance Strategy

Between July 2019 and May 2025, necropsies were performed on 216 wild carnivore carcasses collected across the Valencian Community (Figure 1). The sampled animals included 102 red foxes (Canidae); 29 Eurasian badgers, 31 stone martens, 2 Eurasian otters, and 26 American minks (Mustelidae); 3 European wildcats (Felidae); and 23 common genets (Viverridae).

Most of the 216 specimens included in this study resulted from roadkill (161/216). In addition, sixteen red foxes were obtained through authorised hunting events (independently from our research), and one red fox was found dead of natural causes. Carcasses of protected carnivores were provided by local authorities and the ‘La Granja’ Wildlife Rescue Center, including six stone martens and three common genets that had died from electrocution, as well as three additional common genets that had been illegally trapped. Finally, twenty-six American minks were obtained from official eradication campaigns aimed at controlling this invasive species.

All carcasses underwent a thorough macroscopic examination to assess external and internal lesions. Spleen and blood samples were aseptically collected and stored individually at −20 °C until processing for molecular analysis.

In addition, blood samples were obtained from 34 feral cats (*Felis catus*) as part of a neutering and population control campaign conducted in the municipalities of Sinarcas, Cofrentes, Venta del Moro, and Camporrobles (Valencian Community). Serum was separated and stored at −20 °C until further analysis.

### 2.3. Molecular Detection of Leishmania *spp*.

DNA was extracted from approximately 25 mg of spleen tissue (wild carnivores) or 200 μL of serum (feral cats) using the NZY Tissue gDNA Isolation Kit (NZYtech, Lisboa, Portugal), following the manufacturer’s instructions. DNA was eluted in 60 μL of elution buffer and stored at −20 °C until qPCR analysis. DNA concentration was not measured, and samples were not diluted to equalise DNA concentration prior to qPCR.

Detection of *Leishmania* DNA was performed by amplification of a 139 bp region of kinetoplast minicircle DNA (kDNA) using TaqMan-based quantitative PCR, a method particularly suitable for detecting low parasitic loads in asymptomatic individuals [14,24,25,26].

Each 20 μL qPCR reaction contained NZYSpeedy qPCR Probe Master Mix (2×) with ROX (NZYtech), 8 pmol of each primer (forward: CTTTTC TGGTCCTCCGGGTAGG; reverse: CCACCCGGCCCT ATTTTACACCAA), 4 pmol of the TaqMan probe (FAM-TTTTCGCAGAACGCCCCTACCCGCTAMRA), and 2 μL of sample DNA. Reactions were run on a QuantStudio 5 (Applied Biosystems, Waltham, MA, USA) using 40 cycles with denaturation at 95 °C and annealing/extension at 59 °C.

A no-template control was included in each run as a negative control. A positive control consisting of canine *Leishmania* DNA was kindly provided by the National Reference Center for Leishmaniasis, WOAH Reference Laboratory at the Istituto Zooprofilattico Sperimentale della Sicilia. The DNA was provided at 1 × 10^7^ parasites/mL (equivalent to 1 × 10^4^ parasites/µL) and used to generate a standard curve based on five 10-fold serial dilutions, each tested in duplicate. Although this material had been previously used to validate the analytical sensitivity of a qPCR assay [27], the primers and cycling conditions used here differ, and results are not directly comparable.

Samples were considered positive when they showed a quantification cycle (Cq) value < 37, a Cq confidence ≥ 3.0, and an amplification score ≥ 0.7. The fluorescence threshold was set at 0.2. Borderline or doubtful results were retested for confirmation.

### 2.4. Serological Analysis

Serum samples from 174 wild carnivores were tested for anti-*Leishmania* antibodies using a commercial indirect ELISA kit (VetLine^®^ *Leishmania*, Bio Veto Test, NovaTec Immundiagnostica GmbH, Dietzenbach, Germany), following the manufacturer’s protocol. According to the manufacturer, the diagnostic sensitivity and specificity are both >98%, while predictive values are not reported. The assay is based on *Leishmania* antigens pre-coated on microwells and uses HRP-conjugated Protein A/G as a universal secondary reagent, enabling detection across multiple mammalian species. Optical density (OD) values at 450 nm were measured using a FLUOstar Omega microplate reader (BMG Labtech, Ortenberg, Germany). Results were interpreted according to the manufacturer’s validated cutoff criteria to determine seropositivity. Serum was not available from all sampled individuals, as blood could not be collected in some cases or samples were too degraded to allow reliable separation of serum.

### 2.5. Statistical Analysis

Prevalence values were calculated as the proportion of positive animals over the total tested, and 95% confidence intervals (CI) were determined using the Wilson score method. All CI calculations were performed using the software Quantitative Parasitology 3.0 [28].

Differences in prevalence were evaluated using Fisher’s exact tests (two-sided). Pairwise comparisons were performed for selected taxa (e.g., red fox vs. other species combined; families), and results were interpreted as exploratory due to small counts. For animals tested by both assays, concordance between qPCR and ELISA was assessed with McNemar’s test (two-sided). Analyses were performed in R (v4.3.1); α = 0.05 [29,30].

## 3. Results

A total of 250 individuals, comprising wild carnivores and feral cats and representing eight species, were included in this study. *Leishmania* DNA was detected by qPCR in 14 individuals (Figure 2), yielding an overall prevalence of 5.6% (95% CI: 3.1–9.2%). Positives were detected in five species: red fox (*V. vulpes*), Eurasian badger (*M. meles*), American mink (*N. vison*), stone marten (*M. foina*), and cat (*F. catus*). Red foxes showed the highest prevalence, with 10 out of 102 individuals testing positive (9.8%; 95% CI: 4.8–17.3%). Single positive cases were also recorded in stone marten (3.2%; 95% CI: 1.0–16.7%), badger (3.4%; 95% CI: 1.0–17.8%), American mink (3.8%; 95% CI: 1.0–19.6%), and cat (2.9%; 95% CI: 1.0–15.3%). No positives were identified in common genets (*n* = 23), European wildcats (*n* = 3), or Eurasian otters (*n* = 2). Prevalence values by species, along with their corresponding 95% confidence intervals (Wilson method), are therefore reported directly in the text to reflect the uncertainty associated with each estimate. qPCR prevalence was significantly higher in red foxes than in all other species combined (9.8% (10/102) vs. 2.7% (4/148); Fisher’s exact *p* = 0.023). Family-level comparisons did not reach statistical significance (e.g., Canidae vs. Mustelidae *p* = 0.093), consistent with the small number of positives and the exploratory nature of these analyses.

Macroscopic lesions typically attributed to leishmaniasis were absent in all PCR-positive individuals at necropsy, suggesting a predominantly subclinical infection pattern. An exception was noted in one red fox presenting with severe mange-related dermatitis.

Ct values of PCR-positive animals ranged from 17.46 to 34.83, with the mean Cts observed being 29.94, suggesting low parasite loads. The lowest Ct value (17.46) was observed in a hunted red fox (Vv25016), indicating a high parasitic burden. Table 1 reports the mean Ct values of positive individuals and the estimated parasite concentration per microlitre.

To ensure accuracy in parasite quantification, a standard curve composed of five 10-fold serial dilutions of *L. infantum* DNA was run on the same plate as the field samples. Standards and samples were tested in duplicate. The resulting curve (R^2^ = 0.998, slope = −3.543, efficiency = 91.5%, Figure 3) enabled direct interpolation of Cq values under identical conditions. The estimated parasite concentrations across qPCR-positive individuals ranged from 0.20 to 16,395 parasites/µL (Table 1). Most samples (10 out of 14) showed very low concentrations, below 10 parasites/µL, consistent with subclinical infections. Only three animals presented higher values, and among them, a red fox (Vv25016) showed a markedly elevated load (16,395 parasites/µL). This wide variation suggests that while most infected carnivores likely carried low parasitic burdens compatible with asymptomatic or latent infection, occasional individuals may harbor higher loads that could play a more significant role in parasite transmission.

Serological analysis was conducted on 174 animals, corresponding to those for which serum of sufficient quality was available. Five individuals tested positive by ELISA (their geographical distribution is shown in Figure 4; results are displayed separately for clarity, since not all animals tested by qPCR could also be tested serologically). Among the 11 animals that were qPCR-positive and also tested by ELISA, only one showed concordant positivity in both assays. This individual (a red fox, Vv21001) had an estimated parasite concentration of 56.25 parasites/µL (Ct = 26.2).

The overall seroprevalence was 2.9% (5/174; CI: 95%: 0.9–6.6%). All seropositive individuals were red foxes (*V. vulpes*), yielding a species-specific prevalence of 5.7% (5/88; CI: 1.9–12.8%).

Statistical comparison between the two diagnostic methods did not reveal significant differences (Fisher’s exact test, *p* = 0.235). When restricting the analysis to the 174 animals tested by both assays, McNemar’s test likewise showed no significant difference in detection rates between qPCR and ELISA (*p* = 0.180).

A detailed breakdown of the 250 animals included in the study, including their species, sex, approximate age, sampling year, geographic origin, and diagnostic results, is provided in Appendix A. This table also specifies which individuals underwent serological testing, allowing for a comprehensive overview of the sampling effort and infection status across all tested specimens.

## 4. Discussion

In this study, *Leishmania* DNA was detected in five wild carnivore species from the Valencian Community, with an overall prevalence of 5.6% (CI 95%: 3.1–9.2%). The highest number of PCR-positive cases was observed in red foxes (*V. vulpes*, Canidae), consistent with findings from other studies in southern Europe [9,31,32]. This was also supported by our statistical analysis, which indicated a significantly higher prevalence in foxes compared to all other species combined (Fisher’s exact test, *p* = 0.023), although the limited number of positives suggests this result should be interpreted cautiously. Despite this, parasite DNA was also detected in mustelids, including a badger (*M. meles*), a stone marten (*M. foina*), and an American mink (*N. vison*), as well as in one feral cat (*F. catus*, Felidae), highlighting the diversity of wild and peri-domestic hosts that may harbor *Leishmania* spp. in this endemic region.

In contrast to the low prevalence observed in this study (5.6% by qPCR and 2.9% by ELISA), considerably higher rates of *Leishmania* infection have been reported in wildlife across the Iberian Peninsula and other Mediterranean regions. For instance, Risueño et al. detected *Leishmania* DNA in 44.9% of red foxes sampled in Murcia and parts of the Valencian Community [5]. Similarly, Alcover et al. found prevalence rates of 29.5% by qPCR and 12.8% by ELISA in wild mammals from Catalonia [26], with three individuals testing positive in both assays. Del Río et al. reported molecular prevalence of 28% in wild carnivores from the Basque Country [14]. In central Italy, Verin et al. recorded up to 52.2% PCR-positivity in lymph node samples from red foxes; however, all IFAT-tested sera in that study were negative—a pattern that mirrors our own findings, where molecular detection exceeded serological positivity [33]. Conversely, lower prevalence rates have also been reported in some areas, such as the 5.4% found in foxes by Abbate et al., underscoring the variability across studies [34]. These discrepancies may reflect genuine geographic differences in infection pressure, vector ecology, host population dynamics, or methodological factors including tissue selection, test sensitivity, and sample preservation. Nonetheless, the comparatively low rates observed in this study suggest limited circulation of the parasite among the wild mesocarnivore populations of the Valencian Community at the time of sampling.

The apparent discordance between diagnostic methods also deserves attention. Four individuals were positive by ELISA but negative by qPCR. This can be explained by the temporal dynamics of infection: seropositivity reflects past exposure and the persistence of antibodies even after parasite clearance, whereas qPCR detects active infection at the time of sampling. Similar patterns have been reported in European wildlife. For example, Verin et al. found PCR-positive red foxes that were consistently negative by IFAT, highlighting how molecular detection can exceed serological evidence of exposure [33]. Likewise, Alcover et al. documented partial overlap between qPCR and ELISA results in wild mammals from Catalonia, further supporting that the two techniques often provide complementary rather than overlapping information [26]. In Iberian lynxes, Lima et al. also reported substantial discrepancies between PCR, ELISA, and IFAT results, with only partial agreement among the techniques [12]. Such mismatches may reflect biological factors (e.g., delayed or absent seroconversion, or persistent antibodies after parasite clearance) as well as methodological aspects (e.g., cut-off selection and PCR performance). Overall, these observations reinforce that qPCR is more specific for active infections, whereas ELISA offers insights into past exposure and immune memory, making both approaches complementary tools for wildlife epidemiological surveys.

The majority of PCR-positive individuals presented high Ct values, indicative of low parasite loads. This observation aligns with previous studies reporting similar findings in wildlife [5,35]. Although wild carnivores are likely more exposed to sand fly vectors than domestic dogs or humans, *L. infantum* is primarily adapted to dogs, which are the only confirmed primary reservoir under natural conditions [36]. In wildlife, therefore, infections tend to be incidental: parasite DNA can be detected, but usually at low levels, with inconsistent serological responses and limited clinical impact, supporting their role as spillover hosts rather than effective reservoirs.

Nevertheless, xenodiagnostic experiments have demonstrated that several wildlife species—such as rodents and lagomorphs—are capable of transmitting *Leishmania* spp. to sandflies under controlled conditions [36,37,38]. Moreover, in natural settings, sandflies feeding on wild mammals in outbreak zones have been found to harbor *Leishmania* DNA, as observed during the cutaneous leishmaniasis outbreak in Madrid [38]. These findings indicate that wildlife and peri-domestic species can contribute to parasite transmission under certain conditions. However, their epidemiological relevance as reservoirs in endemic cycles is still uncertain and likely limited when compared to domestic dogs, the main reservoir in the Mediterranean basin.

The only animal that tested positive by both qPCR and ELISA was a red fox with extensive skin lesions affecting over 50% of its body. These lesions were confirmed as sarcoptic mange through skin scraping and microscopic identification of *Sarcoptes scabiei* mites in the laboratory. Interestingly, this individual had one of the lowest Ct values (26.2; 56.25 parasites/µL) among all positive samples, suggesting a relatively high parasite burden. Although a causal relationship cannot be established, we speculate that the *Leishmania* infection may have exacerbated the clinical severity of the sarcoptic mange by further compromising the animal’s immune status. This interpretation remains hypothetical and highlights the need for future studies—including histopathological analyses of affected tissues—to better understand the potential interactions between *Leishmania* and other pathogens in wildlife.

No other PCR-positive animals displayed external lesions. Similarly, in domestic dogs, infection prevalence can exceed 60% in endemic areas, while only a subset of animals develop clinical disease [39]. Thus, parasite detection does not necessarily equate to pathological outcomes in either wildlife or dogs.

Despite detecting *Leishmania* DNA, it is important to note that the PCR assay used targets conserved regions of the kinetoplast minicircle DNA, which prevents the precise identification of *Leishmania* species in the absence of sequencing. Furthermore, although the ELISA test is based on *L. infantum* antigens, it has not been validated in wild carnivores, and potential cross-reactivity cannot be ruled out. Therefore, our findings are conservatively interpreted as exposure to *Leishmania* spp. Nonetheless, as *L. infantum* is the only endemic species in the Mediterranean basin, it is likely the main agent involved.

Another important limitation of this study is the exclusive use of spleen tissue for molecular detection. Although commonly used in wildlife surveillance, this tissue may not always be optimal. As described by Verin et al., thawed spleen samples often become blood-drenched, which can impair molecular detection due to the poor sensitivity of blood for *Leishmania* PCR [33]. Lymph nodes have been shown to offer higher sensitivity in some cases [33,40]. Additionally, the preservation status of carcasses and potential DNA degradation from freezing may have further reduced detection rates.

Although most wildlife species may play a marginal role in maintaining the parasite cycle compared to domestic dogs, surveillance of potential alternative hosts is still essential for One Health strategies. This is particularly important in reintroduction areas of endangered species such as the Iberian lynx (*L. pardinus*), which has been shown to be susceptible to *L. infantum* infection [12]. Understanding the role of sympatric carnivores and the potential risk of transmission is critical to implementing informed conservation measures in endemic regions.

## 5. Conclusions

This study provides new insights into the circulation of *Leishmania* spp. in wild carnivores and feral cats in the Valencian Community, a region endemic for leishmaniasis. The overall low prevalence detected by both qPCR and ELISA, along with the absence of clinical signs in infected individuals, supports the hypothesis that these species may act as part of a sylvatic transmission cycle. However, the detection of parasitic DNA in multiple species highlights the need for continued surveillance, especially given the potential role of wildlife in maintaining the parasite in natural ecosystems. Considering the expanding distribution of *Leishmania* due to climate change, and the close ecological interfaces between wildlife, domestic animals, and humans, integrated monitoring strategies under a One Health framework are essential to better understand and control this zoonotic disease.

## Figures and Tables

**Figure 1 animals-15-02751-f001:**
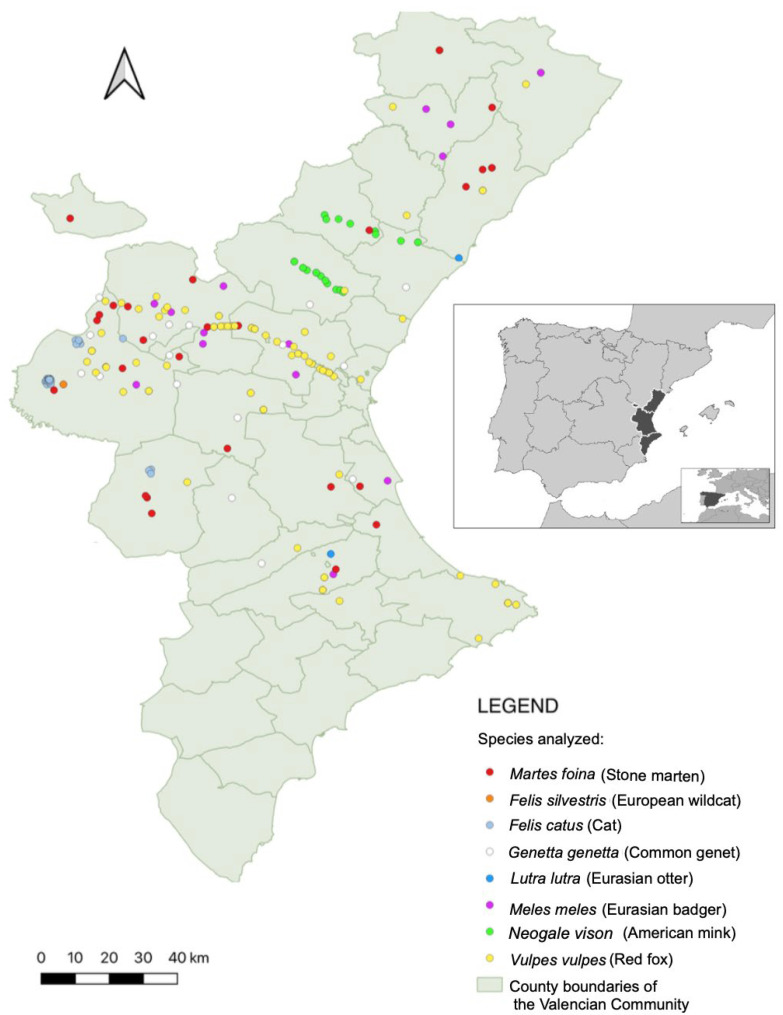
Geographic distribution of wild carnivore and feral cat samples collected across the Valencian Community between 2019 and 2025. Each dot represents an individual tested by qPCR; a subset of these animals also underwent ELISA testing. Each species is represented by a different color, as indicated in the legend. A notable linear concentration of points is visible along the CV-35 motorway, where wildlife-vehicle collisions are frequent. The inset map highlights the location of the Valencian Community within Spain and the Mediterranean basin.

**Figure 2 animals-15-02751-f002:**
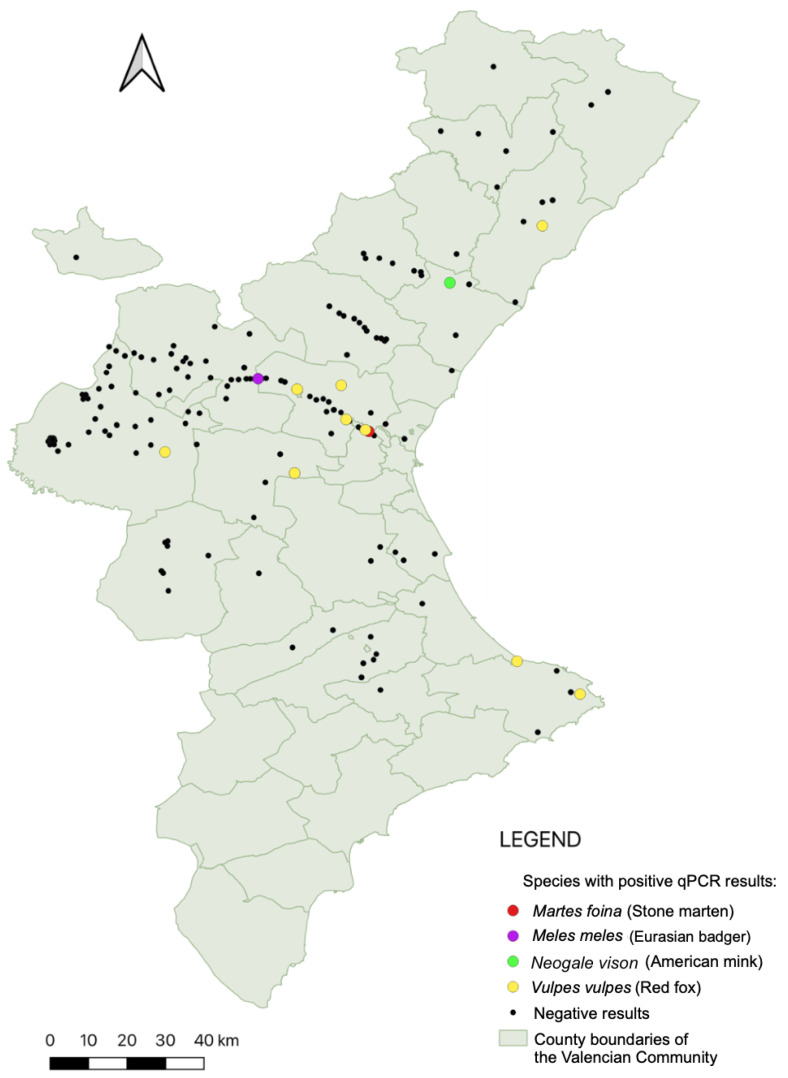
Spatial distribution of qPCR-positive carnivores in the Valencian Community. Colored points indicate individuals that tested positive for *Leishmania* spp. DNA, while black points represent qPCR-negative animals. Species-specific colors are detailed in the legend. Feral cats were excluded from this map to maintain the focus on wildlife species.

**Figure 3 animals-15-02751-f003:**
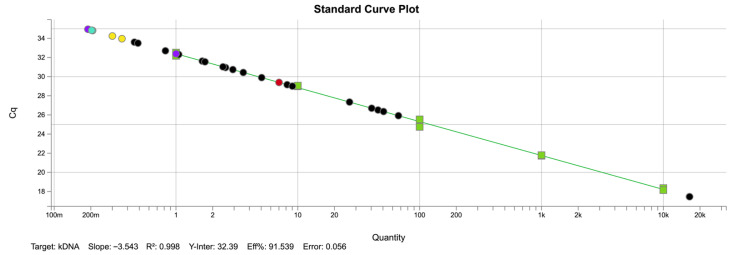
Standard curve generated by TaqMan qPCR for the quantification of *Leishmania* spp. DNA. The curve was constructed using five 10-fold serial dilutions (10^4^ to 1 parasite/µL; green squares) of a positive control sample (1 × 10^7^ parasites/mL), tested in duplicate. The regression line demonstrated high linearity (R^2^ = 0.998), with a slope of −3.543 and an amplification efficiency of 91.5%. The 14 qPCR-positive samples from wild carnivores and feral cats were interpolated against this standard curve to estimate parasite concentrations (parasites/µL). Colored circles indicate the host species: *V. vulpes* (black), *M. meles* (red), *N. vison* (purple), *M. foina* (yellow), and *F. catus* (blue).

**Figure 4 animals-15-02751-f004:**
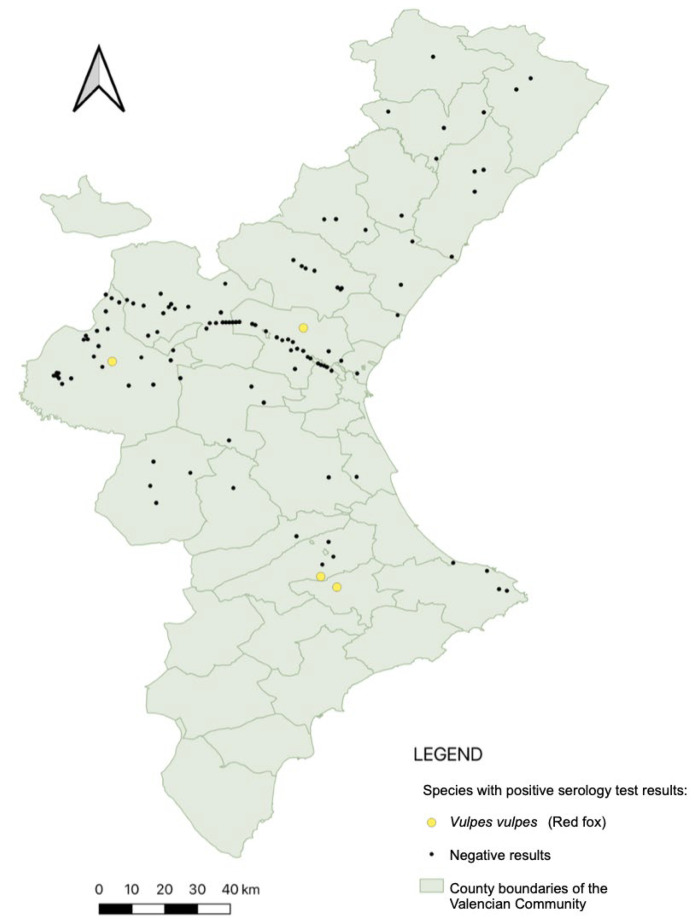
Spatial distribution of ELISA-positive carnivores. Colored points mark individuals seropositive by ELISA; black points represent seronegative animals. See legend for species identification.

**Table 1 animals-15-02751-t001:** Mean Ct values and estimated parasite concentrations in the 14 qPCR-positive individuals.

Sample	Ct Means	Estimated Concentration (Parasites/µL)
Fc23055	34.8295	0.20495
Mf25005	34.1015	0.3302
Mm21005	29.394	7.009
Nv21007	33.671	0.59515
Vv19004	28.2505	17.3735
Vv21001	26.2185	56.25
Vv21023	29.8715	5.976
Vv22002	30.4615	3.7385
Vv24002	31.586	1.6871
Vv24012	30.692	3.0585
Vv24017	32.5095	0.9355
Vv25004	26.527	45.461
Vv25016	17.4575	16,395.0
Vv25019	33.5485	0.47135

## Data Availability

Additional data supporting the findings of this study are available from the corresponding authors upon reasonable request.

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
