# Peer review of "Molecular and Serological Detection of Leishmania spp. in Mediterranean Wild Carnivores and Feral Cats: Implications for Wildlife Health and One Health Surveillance"

_animals, 2025, doi:10.3390/ani15182751_

Round 1

Reviewer 1 Report

Comments and Suggestions for Authors

The study is relevant, however authors should revise carefully, explaining some methodological issues, correct the text to fulfil requirement of the journal and better to provide results obtained, now the number of tables and figures are redundant, some information is repeated.

In supplementary file, there is no name of the Table. Some information is missing, left as blank, for examples for Mm20001 age, sex and origin is missing. In such cases give some definitions, not determined, or?

L38 please indicate examined/infected number of red foxes and percentage, now it is not clear

L85-87 please include Valencian Community

L104 correct to (Figure 1).

L104-107 delete Meles meles and Vulpes vulpes. These Latin names were already mentioned above

L104-107 speices atre listed in chaotic order, please order by families, indicating Canidae, Mustelidae, Felidae and Viverridae families

L110 not clear “one individual was found dead of natural causes” here you talk about foxes, other were hunted? Also 161+16+1 I not equal 216, how other samples were obtained?

L125-130 fix text alignment

L132-135 was the DNA concentration measured, or were the samples diluted to equalise the DNA concentration?

L148 I do not the need of this abbreviation (C.Re.Na.L.)

L171 Figure 2 should be

Figure 2. Why are dots of feral cats not indicated in Figure 2?

L182-183 please rephrase, English should be improved; also Table 1 not table 1.

L188-189 Table has no title, just explanation, every table and figure has to have title.

Table 1 formatting should be done, please delete empty space as one line after the data also blank space is left. Furthermore, the thickness of the borders is too wide than normal

L193 Figure 3, not see figure 3

L197,L226 Leishmania in italic should be

L197 There should be no indentation

Figure 3 it is not clear the choise of colours of circles, why not use the colour as for host species?

L195 “low estimated concentrations across most samples suggest subclinical infections” please provide actual values, the range, what was the low concentrations, and high concentration in which sample(s)? Several sentences here are needed

Table 2 in English use . not ,

L205-209 and Table 2 should move after paragraph ending in L178. Also, from Figure 2 it is clear how many animals were positive for parasite examined; so, it is questionable whether Table 2 is necessary. Important data of the Table 2 is CI, which could be modified as text.

L210 also in Methods not very clear why not on all animals examined, authors should frankly explain the reasoning. I can understand that in the design of study always some obstacles arises

L211 Figure 4

These is no need for separate Figure 4, the information could be included in Figure 2.

L212-213 what was concentration of PCR products in which case?

How authors could explain that four animals were positive by ELISA but negative by more sensitive qPCR? and why didn't the molecular and serological results correlate well with each other? This should be addressed in Discussion

Table 3 is not in appendix but rather presented as supplementary material, authors should correct this and decide there they want to present the detailed raw table. My suggestion not in the same body of manuscript, but rather in supplementary material in that case the tittle also should be presented in text near section Supplementary material.

L228 correct ”5.6 %(CI95%:3.1-9.2%).“ Gaps are missing

L229-232 please provide families of hosts, canids, mustelids and felid

Discussion is this the first detection of Leishmania DNA in American mink in region or worldwide, if no, this host was not provided as previously positive in Introduction L53-60. Correct it

L262 concentration also should be mentioned in Results section

L289 authors could compare if it was significant differences in prevalences estimated by  two different methods used. Please do it and incorporate it into the Results section

Author Response

Comment 1: In supplementary file, there is no name of the Table. Some information is missing, left as blank, for examples for Mm20001 age, sex and origin is missing. In such cases give some definitions, not determined, or?

Response 1: We thank the reviewer for this observation. We have now added a title to the supplementary table (Table S2) and replaced missing values with “Nd,” as clarified in the table legend (Nd = Not determined).

Comment 2: L38 please indicate examined/infected number of red foxes and percentage, now it is not clear

Response 2: Corrected. Numbers and percentages are now included.

Comment 3: L85-87 please include Valencian Community.

Response 3: Corrected.

Comment 4: L104: correct to (Figure 1).

Response 4: Corrected.

Comment 5: L104-107 delete Meles meles and Vulpes vulpes. These Latin names were already mentioned above

Response 5: Corrected.

Comment 6: L104-107 speices atre listed in chaotic order, please order by families, indicating Canidae, Mustelidae, Felidae and Viverridae families

Response 6: Corrected. Species now ordered by Canidae, Mustelidae, Felidae, Viverridae.

Comment 7: L110 not clear “one individual was found dead of natural causes” here you talk about foxes, other were hunted? Also 161+16+1 I not equal 216, how other samples were obtained?

Response 7: We thank the reviewer for pointing out this ambiguity. We agree that the sentence could be misleading. We have rephrased this section to clearly indicate that the 216 specimens included in the study encompassed all animals, and we now provide a step-by-step explanation of their origins and causes of death (page 3, lines 112-119).

Comment 8: L125–130: fix text alignment.

Response 8: Corrected.

Comment 9: L132-135 was the DNA concentration measured, or were the samples diluted to equalise the DNA concentration?.

Response 9: Method clarified.

Comment 10: L148: remove abbreviation (C.Re.Na.L.).

Response 10: Removed.

Comment 11: L171 Figure 2 should be

Response 11: Corrected.

Comment 12: Why are dots of feral cats not indicated in Figure 2?

Response 12: We thank the reviewer for this observation. Our intention was to present Figure 2 as a distribution map of wild carnivores in the Valencian Community. Feral cats were therefore excluded to maintain the focus on wildlife species. This has now been clarified in the figure legend (page 7, lines 216–217)

Comment 13: L182-183 please rephrase, English should be improved; also Table 1 not table 1.

Response 13: Corrected.

Comment 14: L188-189 Table has no title, just explanation, every table and figure has to have title. Table 1 formatting should be done, please delete empty space as one line after the data also blank space is left. Furthermore, the thickness of the borders is too wide than normal

Response 14: Corrected.

Comment 15: L193 Figure 3, not see figure 3

Response 15: Corrected.

Comment 16: L197, L226: Leishmania italic; indentation fix.

Response 16: Corrected.

Comment 17: Figure 3 it is not clear the choise of colours of circles, why not use the colour as for host species?

Response 17: As suggested, we prepared an alternative version of the standard curve figure in which qPCR-positive samples are shown by species (Figure 3).

Comment 18: L195: L195 “low estimated concentrations across most samples suggest subclinical infections” please provide actual values, the range, what was the low concentrations, and high concentration in which sample(s)? Several sentences here are needed.

Response 18: We appreciate this comment and have expanded the text accordingly.

Comment 19: Table 2: English formatting.

Response 19: Table has been removed.

Comment 20: L205-209 and Table 2 should move after paragraph ending in L178. Also, from Figure 2 it is clear how many animals were positive for parasite examined; so, it is questionable whether Table 2 is necessary. Important data of the Table 2 is CI, which could be modified as text.

Response 20: Table removed, CI included in text.

Comment 21: L210 also in Methods not very clear why not on all animals examined, authors should frankly explain the reasoning. I can understand that in the design of study always some obstacles arises

Response 21: We thank the reviewer for this observation. We have clarified this point in the Methods and Results sections. Specifically, we now indicate that serum was not available from all sampled animals because in some cases blood could not be collected, or samples were too degraded to allow reliable serum separation. The Results section was also adjusted to specify that serological analysis was carried out on the 174 individuals for which serum of sufficient quality was available (Methods p. 5, lines 172-174; Results p. 8, lines 244-245).

Comment 22: L211: These is no need for separate Figure 4, the information could be included in Figure 2.

Response 22: We thank the reviewer for this observation. While we initially considered merging Figures 2 and 4, we decided to keep them separate to ensure clarity. Figure 2 presents all animals tested by qPCR, with positive individuals highlighted using species-specific colors. Since not all animals tested by qPCR were also tested by ELISA, combining both datasets into a single figure would have resulted in visual overload and potential confusion. For this reason, we believe that presenting ELISA results separately in Figure 4 provides a clearer and more rigorous representation of the two complementary diagnostic approaches. To address this concern, we have also added a clarifying sentence in the Results section (lines 246-247)

Comment 23: L212-213 what was concentration of PCR products in which case?

Response 23: Corrected.

Comment 24: How authors could explain that four animals were positive by ELISA but negative by more sensitive qPCR? and why didn't the molecular and serological results correlate well with each other? This should be addressed in Discussion

Response 24: We have expanded the Discussion to address this point (page 9, lines 297–313). Specifically, we now explain that ELISA-positive but qPCR-negative cases can be interpreted as evidence of past exposure, where antibodies persist after parasite clearance, whereas qPCR detects active infection at the time of sampling.

Comment 25: Table 3 is not in appendix but rather presented as supplementary material, authors should correct this and decide there they want to present the detailed raw table. My suggestion not in the same body of manuscript, but rather in supplementary material in that case the tittle also should be presented in text near section Supplementary material.

Response 25: Corrected (Table S2).

Comment 26: L228 correct ”5.6 %(CI95%:3.1-9.2%).“ Gaps are missing

Response 26: Corrected.

Comment 27: L229-232 please provide families of hosts, canids, mustelids and felid

Response 27: Corrected.

Comment 28: Discussion is this the first detection of Leishmania DNA in American mink in region or worldwide, if no, this host was not provided as previously positive in Introduction L53-60. Correct it

Response 28: We thank the reviewer for this observation. We had not previously mentioned mink in the Introduction. To correct this, we have now added American mink (N. vison) to the list of wild carnivore species reported as susceptible to Leishmania spp., ensuring consistency with the Discussion (Introduction, page 2, line 58).

Comment 29: L262 concentration also should be mentioned in Results section

Response 29: Corrected.

Comment 30: L289 authors could compare if it was significant differences in prevalences estimated by  two different methods used. Please do it and incorporate it into the Results section

Response 30: We have now compared the prevalence estimates obtained by qPCR and ELISA.  The statistical approach has been added in the Materials and Methods section, and the results have been incorporated at the end of the Results section (Material and Methods, page 5, lines 176–183; Results, page 8, lines 254-257).

Reviewer 2 Report

Comments and Suggestions for Authors

Dear Authors, I’ve reviewed the manuscript titled Molecular and Serological Detection of Leishmania spp. in Mediterranean Wild Carnivores and Feral Cats: Implications for Wildlife Health and One Health Surveillance from Suita et al. While the manuscript is neatly organized and clearly written I believe there are some things that need to be addressed before proceeding with the publication process.

Simple summary:
- Line 25: Could be beneficial to add the species name instead of just addressing the case as “Only one individual”.

Introduction:
- Line 66-68: I’d be cautios in saying that it could be a threat for the conservation of L. pardinus. Is there evidence proving that FeL impacts the health of the species?;
- Line 77: The citation style must be uniformed;
- Line 99: Remove the ( before the citation.

Results
- Line 172 + Figure 2: The correct name of the American mink genus is Neogale;
- Line 197: Leishmania should be formatted in Italics.

Discussion
- Line 237-239-241-242-247-283: The citation style must be uniformed;
- Line 256-258: If wildlife really had higher immunologic efficiency against Leishmania how do you explain the results obtained? Theoretically wildlife is much more exposed to the vector and therefore the parasite, if compared to dogs or humans. But the result show very low seroprevalence, even where parasite DNA was detected;
- Lines 259-267: histopathological analysis of the affected skin would have been beneficial to assess the potential role of Leishmania in exacerbating or participating at all in the described dermatopathy caused by S. scabiei. I believe that without this kind of diagnostic approach, all the paragraph is merely speculative and should be treated as such. Besides, you should add the protocol you use to diagnose S. scabiei infection, could also be a simple skin scrape with microscopic observation, but I believe you need to add this information.
- Lines 288-289: I believe you should’ve performed the statistical analysis. You have a lot of samples and even though you don’t have many that tested positive it is still an interesting scenario that shows that the parasite is not in active circulation in the mesocarnivore community of the sampled area.

References
Some names of genera and species are not formatted in Italics, please go through it again checking for it.

Author Response

We thank the reviewer for the positive evaluation of our manuscript and for the constructive comments. We have carefully revised the manuscript accordingly. Below are our detailed responses to each point raised. 

Simple summary:

Comment 1:
- Line 25: Could be beneficial to add the species name instead of just addressing the case as “Only one individual”.

Response 1: Corrected. Species name specified.

Introduction:
Comment 2: Line 66-68: I’d be cautios in saying that it could be a threat for the conservation of L. pardinus. Is there evidence proving that FeL impacts the health of the species?

Response 2: We thank the reviewer for this important observation. We agree that the clinical impact of Leishmania infantum on Iberian lynx health and conservation is not yet established. We have therefore rephrased the sentence to adopt a more cautious tone, clarifying that although L. infantum has been detected in Iberian lynxes using both molecular and serological methods, its actual consequences for health and conservation remain uncertain and require further longitudinal studies

Comment 3: Line 77: The citation style must be uniformed

Response 3: Corrected.

Comment 4: - Line 99: Remove the ( before the citation..

Response 4: Corrected.

Results

Comment 5: Line 172 + Figure 2: The correct name of the American mink genus is Neogale

Response 5: Corrected.

Comment 6: L197: Leishmania in italics.

Response 6: Corrected.

Discussion

Comment 7: - Line 237-239-241-242-247-283: The citation style must be uniformed

Response 7: Corrected.

Comment 8: - Line 256-258: If wildlife really had higher immunologic efficiency against Leishmania how do you explain the results obtained? Theoretically wildlife is much more exposed to the vector and therefore the parasite, if compared to dogs or humans. But the result show very low seroprevalence, even where parasite DNA was detected

Response 8: We clarified the text to explain that while wild carnivores are more exposed to sand fly vectors, L. infantum is primarily adapted to dogs, which are the only proven primary reservoir. In wildlife, infections are mostly incidental, often controlled by efficient immune responses, resulting in low parasitic loads and limited or inconsistent antibody production. We have modified the text accordingly to improve clarity.

Comment 9: Lines 259-267: histopathological analysis of the affected skin would have been beneficial to assess the potential role of Leishmania in exacerbating or participating at all in the described dermatopathy caused by S. scabiei. I believe that without this kind of diagnostic approach, all the paragraph is merely speculative and should be treated as such. Besides, you should add the protocol you use to diagnose S. scabiei infection, could also be a simple skin scrape with microscopic observation, but I believe you need to add this information.

Response 9: We agree that, without histopathological confirmation, the potential contribution of Leishmania to the dermatopathy must remain speculative. We have revised the discussion to make this explicit, stating that this interpretation is hypothetical and requires further studies to be confirmed. In addition, we clarified the protocol used to diagnose sarcoptic mange: skin scrapings were obtained and examined microscopically, leading to the identification of Sarcoptes scabiei mites. 

Comment 10: Lines 288-289: I believe you should’ve performed the statistical analysis. You have a lot of samples and even though you don’t have many that tested positive it is still an interesting scenario that shows that the parasite is not in active circulation in the mesocarnivore community of the sampled area.

Response 10: We appreciate this suggestion and have now included statistical comparisons. Using Fisher’s exact test, qPCR prevalence in red foxes (10/102; 9.8%) was higher than in all other species combined (4/148; 2.7%; p = 0.023). Exploratory family-level comparisons (Canidae vs Mustelidae, Felidae, Viverridae) did not reach significance (e.g., Canidae vs Mustelidae p = 0.093). In addition, as requested by Reviewer 1, we compared diagnostic methods: overall Fisher’s exact test for qPCR vs ELISA prevalences was not significant (p = 0.235), and McNemar’s test in the 174 animals with both assays was also not significant (p = 0.180). These analyses have been added to the Materials and Methods and Results sections.

References

Comment 11:
Some names of genera and species are not formatted in Italics, please go through it again checking for it.

Response 11: Corrected.

Round 2

Reviewer 1 Report

Comments and Suggestions for Authors

L57-58 “including foxes (V. vulpes), wolves (Canis lupus), badgers (M. meles), lynxes (Lynx spp.), minks (N. vison) and..“ this have to be corrected there are not one species of foxes, wolves, badgers, if authors refer to specific species, they should indicate this. Introduction is separate part from Abstract and Simple summary, so at first species mentioning full English and Latin names has to be used; red foxes (Vulpes vulpes), grey wolves (Canis lupus), Eurasian badgers (Meles meles), lynxes (Lynx spp.), American mink (Neogale vison) …

L108-111 Family names has to be in normal style, not italic. …102 red foxes (Canidae) 29 Eurasian badgers, 31 stone martens (Martes foina) 2 Eurasian otters (Lutra lutra), and 26 American minks (Mustelidae); 3 European wildcats (Felis silvestris; Felidae); and 23 common genets (Genetta genetta; Viverridae).

L123  change to: “from 34 feral cats (Felis catus)”

L179-183 references should be included

L198-199 Canidae, Mustelidae – normal style

L196-197 “…qPCR prevalence was significantly higher in red foxes than...”

Author Response

General reply:
We sincerely thank the reviewer for the helpful comments. Below we provide our responses and indicate the corresponding corrections made in the manuscript.

Comment 1 (L57–58): “including foxes (V. vulpes), wolves (Canis lupus), badgers (M. meles), lynxes (Lynx spp.), minks (N. vison) and…” — this has to be corrected. There are not one species of foxes, wolves, or badgers; if authors refer to specific species, they should indicate this. The Introduction is a separate part from the Abstract and Simple Summary, so at first mention, full English and Latin names have to be used; red foxes (Vulpes vulpes), grey wolves (Canis lupus), Eurasian badgers (Meles meles), lynxes (Lynx spp.), American mink (Neogale vison)…

Reply 1: We thank the Editor for this observation. The sentence has been corrected as requested to specify the full English and Latin names at first mention:
"…including red foxes (Vulpes vulpes), grey wolves (Canis lupus), Eurasian badgers (Meles meles), lynxes (Lynx spp.), American minks (Neogale vison) and martens (Martes spp.)…"
Regarding Martes spp., in this case we are intentionally referring to the genus Martes, since several species within this genus have been described as potential hosts in the literature. Therefore, we have kept Martes spp. rather than restricting it to a single species. We hope this clarification addresses the concern.

Comment 2 (L108–111): Family names have to be in normal style, not italic.
Reply 2: Corrected.

Comment 3 (L123): Change to: “from 34 feral cats (Felis catus)”.
Reply 3: Corrected.

Comment 4 (L179–183): References should be included.
Reply 4: We thank the reviewer for this suggestion. We have now included two additional references in this section to support the statement.

Comment 5 (L198–199): Canidae, Mustelidae – normal style.
Reply 5: Corrected.

Comment 6 (L196–197): “…qPCR prevalence was significantly higher in red foxes than…”.
Reply 6: The word “significantly” has been added. We thank the reviewer for this helpful comment.